# Fluorescence Molecular Targeting of Colon Cancer to Visualize the Invisible

**DOI:** 10.3390/cells11020249

**Published:** 2022-01-12

**Authors:** Thinzar M. Lwin, Michael A. Turner, Siamak Amirfakhri, Hiroto Nishino, Robert M. Hoffman, Michael Bouvet

**Affiliations:** 1Department of Surgery, University of California San Diego, San Diego, CA 92093, USA; thinzar_lwin@dfci.harvard.edu (T.M.L.); maturner@health.ucsd.edu (M.A.T.); siamirfakhri@health.ucsd.edu (S.A.); hnishino@health.ucsd.edu (H.N.); meishale@gmail.com (R.M.H.); 2Department of Surgical Oncology, Dana Farber Cancer Institute, Boston, MA 02215, USA; 3VA San Diego Healthcare System, San Diego, CA 92161, USA; 4AntiCancer, Inc., San Diego, CA 92111, USA

**Keywords:** tumor-specific antibodies, fluorescent dyes, tumor-specific labeling, fluorescence-guided surgery

## Abstract

Colorectal cancer (CRC) is a common cause of cancer and cancer-related death. Surgery is the only curative modality. Fluorescence-enhanced visualization of CRC with targeted fluorescent probes that can delineate boundaries and target tumor-specific biomarkers can increase rates of curative resection. Approaches to enhancing visualization of the tumor-to-normal tissue interface are active areas of investigation. Nonspecific dyes are the most-used approach, but tumor-specific targeting agents are progressing in clinical trials. The present narrative review describes the principles of fluorescence targeting of CRC for diagnosis and fluorescence-guided surgery with molecular biomarkers for preclinical or clinical evaluation.

## 1. Introduction

Colorectal cancer (CRC) is a common cause of cancer and cancer-related death. Surgery is the only curative modality. Traditional approaches to localize and confirm complete resection during surgery are: preoperative cross-sectional imaging, identification of anatomic boundaries, palpation of the lesion, clinical judgment, and frozen sections if the degree of suspicion is high. However, this traditional approach can be subject to detection and sampling errors, which can leave behind positive margins. Traditional approaches become even more challenging in the setting of neoadjuvant chemotherapy and radiation, as scarring and fibrosis can mask viable tumors. Fluorescence imaging can enhance the delineation of the tumor-to-normal tissue interface. Tumor-specific antibodies tagged with fluorophores can provide real-time in situ imaging. Adjuncts to traditional approaches during surgery can further enhance detection and potentially increase rates of curative resection. The present narrative review describes approaches to the selective targeting of CRC with fluorescent probes specific for tumor biomarkers. 

## 2. Impact of Margin Positivity on Colorectal Cancer and the Potential of Fluorescence Guidance

With 1.9 million new cases of colorectal cancer in 2020 causing 940,000 deaths, CRC is the second most common cause of cancer-related deaths worldwide [1]. In the United States, it is the third most common cause of cancer and cancer-related death [2]. While advances in systemic therapies have improved outcomes for patients with advanced colorectal cancer, surgery remains the mainstay of curative-intent treatment in early stages. 

Complete resection with negative margins is the primary aim of oncologic surgery. For colon cancer, this involves removal of the colon segment containing the tumor, its affected vascular pedicle, and the lymphatic drainage basin of the affected segment [3]. However, this remains challenging, as the standard intraoperative methods of visual inspection and palpation to assess margin negativity are highly subjective. Further assessments via intraoperative pathologic frozen sections are time-consuming and can be affected by sampling error, as there is a large surface area to sample from. This can be further exacerbated by the trend towards minimally invasive resections for colorectal cancer and the trend towards total neoadjuvant therapy. While the minimally invasive approach can enhance visualization with magnification, it is a narrower field of view, and the approach limits the traditional approach of palpation and assessment relies primarily on visual cues. With neoadjuvant therapies, areas of scarring and fibrosis can be indistinguishable from cancer. Viewing with the naked eye under white light is often insufficient to detect contrast between cancer and normal tissue, especially because tumors can often have similar color and texture to those of adjacent tissue. 

Positive margins and early recurrences indicate that residual tumor was left behind at the time of surgery. In a population analysis of 1.2 million patients from the National Cancer Database, Orosco et al. reported that up to 83,241 patients (6.8%) had positive surgical margins [4]. The rate is even higher for rectal cancer. Even after adopting the total mesorectal excision (TME) approach along with effective multimodal therapies, the rates of positive margins were still range from 17–22% [5,6,7]. Patients with positive surgical margins had a local recurrence rate of 22%, compared with 4% of those with negative margins [8]. In a large meta-analysis examining over 17,000 patients with rectal cancer, Quirke et al. showed that positive or close circumferential margins (less than 1 mm) were strong predictors of local recurrence (HR 6.3, 95% CI 3.7–16.7 with no neoadjuvant therapy and HR 2.0, 95% CI 1.4–2.9), distant metastases (HR 2.8, 95% CI 1.9–4.3), and survival (HR 1.7, 95% CI 1.3–2.3) [9]. Additionally, when neoadjuvant therapy was used, positive margins had an even greater association with recurrence (HR 6.3, 95% CI 3.7–16.7), indicating the lack of reliability using bright-light visualization and palpation in a pretreated field. 

In metastatic CRC, margins matter even more. Liver metastases from CRC (CLM) are seen in one-third of patients, and margin-negative resection is still the only potentially curative option [10]. Vandeweyer et al. showed that in patients with stage 4 CRC with isolated liver metastases amenable to resection, there was a difference in the 5-year overall survival of 25% vs. 43% (*p* < 0.04) when resection margins were less than 1 mm compared to resection margins greater than 1 mm [11]. 

Fluorescence labeling of tumors allows surgeons to better delineate anatomic structures in real time, as there is immediate feedback while the tissue is being manipulated. Fluorescence-guided surgery (FGS) does not use ionizing radiation or bulky specialized equipment (i.e., intraoperative ultrasound, CT or MRI), and has direct visual concordance with the surgical field. In oncologic surgeries, a targeted probe can provide additional information on the molecular characteristics of the tumor. FGS can potentially be used during all colorectal surgeries, especially for deep or locally-aggressive colon cancers, rectal cancers, cancers after neoadjuvant treatment, and cancers where an organ-sparing approach is being utilized (i.e., submucosal or full-thickness resections). The technology can be focused on the primary tumor, lymph node spread, detection of peritoneal disease, liver metastases, surveillance of the resection bed, or identifying areas where further microscopic pathologic analysis is necessary. 

Intraoperatively, fluorescence imaging can be performed as a diagnostic laparoscopy to detect the primary tumor and or peritoneal/liver metastases. Once the surgery is underway, fluorescence imaging can be utilized periodically to better define the location of the tumor in relation to surrounding tissue and anatomic structures. This can be performed in a minimally invasive fashion with fluorescence-capable laparoscopes or in a traditional open laparotomy with hand-held devices. Fluorescence imaging can potentially be used to detect lymphatic drainage, although this can be impaired due to the depth of overlying tissue in patients with obesity and thickened fatty mesentery. After the specimen is removed, fluorescence imaging can be performed on both the surgical bed and the specimen to ensure complete tumor removal and an adequate rim of normal tissue or mesorectal envelope. 

Fluorescence labeling to improve cancer detection can also be applicable to endoscopy, the primary prevention strategy for reducing overall mortality from CRC. Along with the increased availability of fluorescence-enabled laparoscopes, endoscopes with fluorescence capabilities are being developed. Similar to surgery, endoscopy faces the challenges of detecting a lesion from a large surface area and limited contrast with standard white light. Tandem colonoscopy, a method in which two same-day colonoscopies are performed on a patient, is the most reliable approach for investigating the adenoma miss rate (AMR). Tandem studies have shown that the miss rate of total polyps is as high as 22%, indicating an additional field in which fluorescence can improve disease detection in CRC [12]. Fluorescence can improve the detection rate of CRC from adenomas and provide further information to care for patients in whom a watch-and-wait approach is being used after neoadjuvant treatment.

## 3. Principles of Fluorescence and Intraoperative Fluorescence Imaging 

Fluorescence occurs when a molecule that has absorbed light of a shorter wavelength (higher photon energy) and emits light of a longer wavelength (lower photon energy) [13]. Fluorescence can occur endogenously or be administered exogenously with a fluorescent dye or targeted probe.

Endogenous fluorescence naturally occurs within the tissue by molecules such as heme, porphyrins, NADH, FAD, collagen, or elastin. These molecules emit signals in the visible spectrum (400–700 nm) [14]. Attempts to distinguish between tumor and normal tissue using endogenous tissue autofluorescence have been successful in parathyroid surgery, and this is an active area of study in fluorescence-guided endocrine surgery [15]. With regard to CRC and other solid GI malignancies, endogenous fluorescence signals for tissue discrimination between normal tissue and cancer have been explored, but have not been incorporated into routine clinical use [14].

Administration of exogenous fluorescence contrast can further enhance the difference between abnormal tissue such that the signal is feasible for use in surgical navigation. When using exogenous fluorescence agents, a near-infrared wavelength (NIR) in the 700–900 nm range has been preferred since overlapping nonspecific fluorescence from endogenous fluorophores is eliminated and light scattering or tissue quenching of the desired signal is decreased [16]. Using wavelengths in the NIR range allows for increased tissue depth penetration and an improved signal-to-noise ratio compared to fluorophores in the visible spectrum [17]. 

## 4. Non-Targeted Fluorescence Agents

Exogenous fluorescence agents can be targeted or non-targeted. Non-targeted NIR agents include dyes such as 5-aminolevulinic acid (5-ALA), methylene blue, and indocyanine green (ICG), all of which are FDA approved. It is hypothesized that there is preferential tumor accumulation of nonspecific dyes through an enhanced permeability and retention effect [18,19]. These dyes are localized and retained in tumors due to defects in their endothelium rather than tumor-specific ligand–receptor-mediated mechanisms [20]. 

5-ALA is a semi-selective dye, as its metabolite, protoporphyrin IX, is preferentially retained in cancer cells due to deficient ferrochelatase activity [21]. It is orally administered and has been widely used in neurosurgery [22]. It was evaluated during diagnostic laparoscopy to detect peritoneal disease by Kondo et al., and a fluorescence signal was seen in peritoneal surface lesions in 8 of 12 patients (66.7%) [23]. 5-ALA was evaluated in a larger trial for the identification of metastatic CRC in lymph nodes during surgery, but it had limited ability to demonstrate fluorescence in the primary tumor (35.3% in cohort 1 and 33.3% in cohort 2) with poor sensitivity and limited specificity for the detection of lymph nodes (sensitivity cohort 1 = 11.1%, cohort 2 = 0%) (specificity cohort 1 = 75%, cohort 2 = 75%) [24]. 

Methylene blue is commonly used as a visually-detectable dye for sentinel lymph node detection [25], but when delivered at lower doses and imaged with an NIR system, methylene blue emits fluorescence in the 660 nm range. As the molecule is renally eliminated, it has been used to image ureters during colorectal surgeries [26,27]. As a direct tumor-labeling agent, it has been explored for use in breast cancer but not for the direct labeling of CRC [28,29]. While it is beyond the scope of the present review, chromoendoscopy using methylene blue to enhance colonic polyp detection is a field under active investigation in the diagnostic field. 

Indocyanine green (ICG) is a nonspecific, hepatically-metabolized dye that was initially developed in the 1950s for retinal angiography, cardiac function, and hepatic function testing [30]. It is now the most commonly used molecule in the field of fluorescence-guided surgery and has gained widespread popularity for a wide variety of NIR fluorescence applications. ICG has been used at varying doses and schedules to image a wide array of anatomic structures: the biliary tree, soft tissue flaps, blood vessels, ureters, lymphatic vessels, lymph nodes, and tumors [31]. Within colorectal surgery, ICG has most frequently been used to evaluate the perfusion of anastomoses, and meta-analyses showed that the use of fluorescence angiography decreased anastomotic leaks, especially in lower rectal anastomoses [32,33,34]. 

ICG is less commonly used to label tumors since a specific molecular target is preferred, but several studies have evaluated ICG tumor-labeling. ICG is FDA approved, in contrast to other molecularly targeted agents, which are only available through clinical trials. Several Japanese groups have reported case series with direct endoscopic injection of ICG in the CRC prior to intraoperative imaging, with reasonable success at visualizing the tumor [35,36,37]. The limitations of this approach, beyond the need for a separate endoscopy for injection, are that mucosal injections of ICG may not be visible from the serosal side during surgery, especially if there is a high degree of intraabdominal obesity, and that there tends to be local diffusion of the molecule and a lack of contrast between the surrounding tissue and the primary lesion. 

Systemic administration of ICG over time leads to preferential ICG accumulation in not just the primary tumor but also any peritoneal metastases. Cao et al. evaluated a series of 11 patients undergoing surgery for CRC who received 25 mg of ICG intravenously, and they evaluated the fluorescence signal at the tumor at a variety of time points [38]. There were 10/11 (91%) tumors with positive fluorescence and an optimal tumor-to-background ratio (TBR) of 1.9–2.2 at 2–4 h after injection. There was fluorescence in 38/40 (95%) lymph nodes evaluated. In one patient, additional nodules were detected only by ICG fluorescence. ICG has been used to evaluate peritoneal carcinomatosis from CRC by three groups. Liberale and Lieto et al. both used a weight-based dose of 0.25 mg/kg of ICG during surgery and showed fluorescence positivity in malignant nodules with a good sensitivity of 87.5–96.9% and specificity of 75–100% [39,40]. Barabino et al. used a similar dose in patients with peritoneal carcinomatosis 24 h prior to surgery and found lower sensitivity and specificity of 72.4% and 60.0%, respectively which suggests that 24 h after a lower dose of ICG may be insufficient for optimal fluorescence enhancement. This is in comparison to the “second window” use of ICG, where larger amounts of the dye (5 mg/kg) are administered intravenously and imaged after 24 h. This approach has been useful for delineating a number of tumors including tumors in the pancreas [41], lung [41,42], brain [43,44], and prostate [45]. High dose ICG labeling was reported as feasible in one case report of a patient with pulmonary CRC metastasis, but this approach has not been used for direct labeling of CRC [46]. 

## 5. Targeted Fluorescence Agents

While non-targeted dyes are easily accessible because they are FDA approved, tumor-specific agents are appealing due to their ability to tailor the probe to specific targets and their ability to provide additional information on the molecular characteristics of the tumor. Molecular specificity can be conferred through several different classes of tumor-targeting probes such as antibodies, antibody fragments, and nanobodies (Figure 1) [47]. Other probes include protein scaffolds, aptamers, peptides, small molecules, and nanoparticles.

Antibodies are the traditional and most commonly used approach for tumor targeting since they have been developed against a wide variety of targets, are easily modifiable, and can be produced in large amounts. Antibodies are the focus of a long and broad body of literature, including clinical studies for drug delivery and radionuclide studies [49]. They can easily be conjugated to fluorophores to act as molecular tracers for intraoperative imaging. Antibodies can be fragmented into smaller molecules that maintain their antigen-binding properties, such as diabodies or minibodies, but these have not gained popularity for clinical use, as the fragments were unstable and difficult to optimize for larger production [50,51,52]. Smaller molecules can be advantageous, as they are able to provide higher contrast at earlier time points compared to intact antibodies. This avoids the need for patients to have a separate clinical visit to receive the tracers, and the molecule can be administered on the same day of surgery. However, the disadvantage with smaller molecules is that they can have a decreased overall tumor signal due to rapid renal elimination and the decreased number of overall fluorophores that can be conjugated. Molecules smaller than 60 kDa are renally filtered, resulting in more rapid blood clearance and leaving decreased time for target binding. A newer class of biological targeting molecules for FGS are the monomeric antigen-binding domains of heavy-chain-only antibodies, also known as single-domain antibodies (sdAbs) or nanobodies [53]. They are advantageous because they can be produced in animal, yeast, or bacterial cells and have high thermal and chemical stability for molecular conjugations and are promising for FGS [54]. 

Non-antibody-based platforms for conferring tumor specificity include protein scaffolds. This class of molecules, also known as antibody mimetics, can bind antigens but are not structurally related to antibodies [55]. Molecules in this class have similar backbone structures, have high protein folding stability, and can be easily and cost-effectively produced in bacterial hosts. They include molecules such as affibodies, adnectins, DARPins (designed ankyrin repeat proteins), centyrins, and knottins, among many others [56]. Identification of binding partners occurs via high-throughput screening against a peptide library [57]. While many are being actively explored for therapeutic uses, relatively few studies have been performed for intraoperative molecular imaging. Other drawbacks are the same as those of other small molecules: rapid renal clearance and limited fluorophore conjugation. Antibody mimetics have promising potential as therapeutic agents, and there is increasing interest in using these molecules as probes for FGS [58]. 

Aptamers are single-strand nucleotide-based molecules with a three-dimensional structure that enables specific target binding. Similar to protein scaffolds, binding partners are identified through an iterative selection process termed systematic evolution of ligands by exponential enrichment (SELEX) [59]. These molecules are generated with efficiency using a solid-phase oligonucleotide synthesis, have excellent biochemical stability, and are internalized by target cells. The limitations of aptamers are similar to those of other small molecules, but an additional major limitation is that aptamers are degraded in-vivo by endogenous nucleases and immune detection by toll-like receptors [60]. Chemical modifications are being investigated to overcome these limitations, and aptamers are also beginning to be investigated for FGS. 

Other classes of tumor-targeting molecules are peptides and nanoparticles. These molecules form a heterogeneous group, as their structures and biochemical properties are variable, and they do not share a common backbone. These include protease-activatable peptides that contain a quencher that is cleaved off upon activation with proteases, which are upregulated in tumors [61]. Other peptides bind to naturally occurring motifs on molecules upregulated in cancer, such as integrins or folate receptors [47]. Nanoparticles are very small molecules, usually 1–100 nanometers in diameter, with efficient cell penetration. They have highly versatile structures for encapsulating fluorescent molecules and can be linked with tumor-targeting moieties for FGS [62]. 

## 6. Molecular Targets for CRC

There are a large number of molecular targets suitable for therapeutic targeting of CRC and drug development, but biomarkers appropriate for tumor-specific imaging have slightly different characteristics. 

Biomarkers suitable for tumor targeting were systematically described by van Oosten et al. in 2011 by the Target Selection Criteria (TASC) scoring system [63]. Potential molecular targets are ranked for tumor-specific imaging by the following characteristics, which make them ideal for FGS. The most important features are as follows: (1) The target must be present on the cell surface or be in close proximity to the tumor cell, making it accessible to the fluorescent probe. (2) The target must be upregulated by most cancer cells within the tumor. (3) Expression of the target should be minimal in normal tissue such that the tumor-to-normal ratio is greater than 10. (4) The target is present at a high frequency in the population of patients with CRC. Other characteristics, such as the previous use of this biomarker for in vivo imaging studies, the presence of enzymatic activity in and around the tumor, and the presence of internalization of the probe-target complex, are features that were also ranked. 

Using this system, the group identified the following targets for optimal development in FGS of CRC: human carcinoembryonic antigen (CEA), CXC chemokine receptor 4 (CXCR4), epidermal growth factor receptor (EGFR), epithelial cell adhesion molecule (EpCAM), matrix metalloproteases 1, 2, 3, 7, and 9 (MMPs), Muc 1, and vascular epithelial growth factor-A (VEGF-A). Targets such as CEA, EGFR, EpCAM, MUC1, and VEGF were also highlighted as promising tumor-associated targets by the National Cancer Institute [64]. The present review describes the in-vivo preclinical and clinical status of the targets identified above. Targets that do not necessarily meet the optimal TASC scoring criteria but are currently under active exploration for surgical applications in CRC are also discussed in the present review.

## 7. Human Carcinoembryonic Antigen

Human carcinoembryonic antigen (CEA) is one of the most well-studied tumor biomarkers in CRC [65]. It is a membrane-bound glycoprotein involved in cell adhesion, expressed during embryogenesis, with little to no expression in normal tissue, and is highly expressed in patients with CRC (>90%) [66,67]. CEA is used as a serum tumor marker and expression has been linked to prognosis [68]. CEA is expressed at high levels in tumors and is shed into the serum, although serum levels of CEA do not necessarily reflect tumor expression of CEA [69,70,71]. There has been a number of preclinical and clinical studies using CEA antibodies as probes for FGS. Tumor targeting using an anti-CEA antibody was demonstrated to highlight colorectal cancer and improve rates of recurrence and overall survival in mouse models of CRC [72,73,74]. Other studies using fragmented CEA antibodies for radionuclide studies have been undertaken, but the use of an intact anti-CEA antibody has advanced the furthest in clinical studies [75]. SGM-101 is an anti-CEA antibody conjugated to a 700 nm fluorophore developed by Surgimab. This molecule showed safety and efficacy in phase I/II clinical trials for resection of CRC and peritoneal metastases from CRC [76,77,78]. There were 12/37 patients who had their surgical plans altered due to fluorescence molecular labeling of tumors with SGM101. It is noteworthy that with a high negative predictive value (NPV) (94%) and sensitivity (96%) at the optimal dose of 10 mg, it can potentially be utilized for organ salvage and watch-and-wait approaches 7/7 patients without fluorescence in the tumor after injection of the probe had pathologically complete responses from neoadjuvant therapy (Figure 2). SGM-101 is currently undergoing a phase III multicentered international clinical trial for intraoperative delineation of primary, recurrent, and metastatic CRC (NCT03659448). 

Other preclinical platforms to target CEA include the use of nanobodies, antibody-linked nanoparticles, and the targeting of multiple members of the CEA family. As described above, nanobodies are smaller than antibodies (~15 kDa vs. ~150 kDa) but retain the specificity of antibodies and reach peak binding of the target within a shorter time frame [79]. An anti-CEA nanobody conjugated to an 800 nm fluorophore was able to delineate subcentimeter colon cancer tumors in mouse models within 1–3 h [80]. Nanoparticles are another promising class of tumor-targeting molecules that are unique in that they can potentially overcome many of the traditional limitations of traditional antibody–fluorophore conjugation, such as improved emission brightness, quantum yield, and photostability [81]. Silica nanoparticles linked to anti-CEA antibodies have been evaluated in CRC mouse tumor models and showed tumor-specific localization after 6 h [82]. Targeting other molecular targets in the CEA family such as carcinoembryonic antigen-related cell adhesion molecules (CEACAMs), can potentially be useful for in-vivo labeling. CEACAMs play a role in cell signaling, adhesion, and tumorigenesis in CRC [83,84]. As there may be differential CECAM expression in CRC, targeting multiple CEACAMs could enhance detection if one or more CEACAMs are not overexpressed [83]. The use of CEA for tumor targeting is also applicable to pancreatic cancer as well as CRC [85,86,87]. There are promising results in both preclinical and clinical studies on multiple targeting platforms targeting CEA. CEA and CEA-related adhesion molecules are appealing biomarkers to target for FGS of GI malignancies. 

## 8. CXCR4

CXC chemokine receptor 4 (CXCR4) is a transmembrane chemokine receptor that binds chemokines CXCR 7 and 12. Under normal conditions, CXCR4 functions in bone marrow and hemopoietic cell signaling [88]. It is also overexpressed in a number of different cancers, including CRC [89,90]. In CRC, CXCR4 overexpression is a negative prognostic indicator, and CXCR4 expression has a positive correlation with the TMN stage, lymph node involvement, and the rate of metastasis [89,91]. CXCR4 is overexpressed in 60–70% of CRCs [92]. While there has been some interest in using CXCR4 for radionuclide targeting and some in vivo fluorescence labeling of GI cancers, it is limited by off-target binding to white blood cells and accumulation in normal CXCR4-expressing organs such as bone marrow and spleen. Unzueta et al. used a cell-penetrating peptide linked to a green fluorescent protein (GFP) to target CXCR4 in a metastatic CRC mouse model, demonstrating proof-of-principle and the potential to use CXCR4 to target tumors. CXCR4 labeling has been shown to be feasible with small molecules and peptides linked to fluorophores in breast, brain, and bladder cancers [93,94]. However, there have been no clinical trials targeting CXCR4 for FGS despite its high score in TASC criteria [63]. 

## 9. EGFR

Epidermal growth factor receptor (EGFR) is a transmembrane tyrosine kinase receptor expressed during embryonic cell development and maintains epithelial cell homeostasis in adult tissue [95,96]. EGFR overexpression is common in many tumors, and it is estimated to be overexpressed in 60–80% of CRCs and is associated with a poor prognosis [97]. Inhibitors of EGFR were among the first targeted drugs for cancer and currently constitute the standard of care for lung and CRC patients [98]. EGFR is an attractive therapeutic and diagnostic target, and many small molecule inhibitors and antibodies are already FDA approved or undergoing active clinical trials in a number of cancers [99]. It is also an appealing target for tumor imaging, and EGFR antibodies and antibody fragments have been the subject of radionuclide studies [100,101]. In the field of FGS, an anti-EGFR antibody (either chimeric cetuximab or humanized panitumumab) conjugated to an 800 nm fluorophore has been the most extensively studied molecule. Marston et al. studied panitumumab-IRDye800 in mice with subcutaneous CRC and found that the molecule resulted in a significantly higher TBR in all three models compared to a nonspecific IgG-IRDye800 [102]. This conjugate has been well evaluated in preclinical and clinical studies of head and neck cancer [103,104,105,106], pancreatic cancer [107,108,109], and brain cancer [110,111]. A study using cetuximab-IRDye800 for locally advanced rectal cancer (TRACT-II) is ongoing in the Netherlands (NCT04638036). Other fluorophore–antibody conjugates undergoing evaluation include nimotuzumab, a humanized EGFR antibody [112]. Alternative probes targeting EGFR include the EGFR affibody (ABY-029), which has completed phase I clinical trials for gliomas, sarcomas, and head and neck cancers (NCT02901925, NCT03154411, NCT03282461) [113], and EGFR nanobodies which have demonstrated tumor labeling in mouse models of head and neck and cervical cancer [114]. As there are a number of molecules that have been developed to bind and inhibit EGFR, along with applicability to multiple different tumor types, EGFR should be an appealing target for FGS [115]. 

## 10. EpCAM

Epithelial cell adhesion molecule (EpCAM) is a transmembrane glycoprotein expressed during embryonic development and is overexpressed in many epithelial carcinomas, especially CRC. EpCAM overexpression was noted in 80–90% of CRCs [116,117]. However, the function of EpCAM in CRC is complex, as high EpCAM expression is associated with poor prognosis in primary CRC and improved prognosis in CRC metastases [118]. Given the complexity of EpCAM expression and function, probes to target EpCAM are relatively less developed. As such, fluorescent molecular targeting of CRC using EpCAM are in preclinical phases. Van Driel et al. studied an EpCAM-targeting antibody to label mouse CRC as well as CRC peritoneal carcinomatosis [119]. Boogerd et al. used an antibody fragment targeting EpCAM conjugated to IRDye800 to evaluate labeling of colon cancers in mouse models [120]. This antibody fragment is being developed for clinical translation. With strong expression and labeling in not just CRC but also other epithelial cancers, further clinical evaluation of EpCAM for FGS will be forthcoming. 

## 11. Matrix Metalloproteases

Matrix metalloproteinases (MMPs) are calcium-dependent zinc-containing endopeptidases that function to degrade the extracellular matrix and have a wide variety of physiological functions, especially in cell proliferation and migration [121]. Overexpression of MMP-1, -2, -3, -7, -9, and -13 and MT1-MMP has been demonstrated at rates of 30–90% in CRCs, depending on the type of MMP being evaluated [122,123]. Probes targeting MMPs for labeling tumors and FGS are usually designed as activatable peptides: synthesized with a protease-cleavable linker and a fluorophore that is activated upon cleavage [124]. They are usually small (several kDa), with rapid pharmacokinetics and improved tissue penetration, and can be either topically applied or intravenously administered. Because these molecules are activated by enzymatic cleavage, one enzyme can cleave multiple molecules of the probe, and the fluorescence can be locally amplified. This approach decreases off-target signals. The drawbacks are that these probes can potentially be activated in the serum, as there are endogenous metalloproteinases, and that these small molecules can diffuse away from the tumor without active cell penetration or uptake [31].

Weissleder et al. described imaging of protease activity in vivo [125] and using this technology, developed a series of protease-cleavable fluorophore-conjugated molecules. MMPSense 680 is a molecule that contains a quenched NIR fluorophore that fluoresces after cleavage of the quencher. MMPSense 680 is a probe cleaved by a broad array of MMPs, especially MMP-7 and -9, and was demonstrated to detect spontaneously developing early colorectal adenomas in mouse models by Clapper et al. [126]. 

Nguyen et al. developed ratiometric activatable cell-penetrating peptides (ACPPs) that enter cells after MMP-2 and -9 cleavage [127]. The molecule contains two fluorophores; protease-mediated degradation of the linker leads to a local increase in one fluorophore over the other, and the ratiometric difference is detected rather than absolute fluorescence, which increases the specificity of the signal. An ACPP called AVB-620, has completed phase I/II trials in patients with breast cancer [128,129]. While AVB-620 has not been evaluated in CRC, Zeng et al. described imaging of mouse models of CRC and metastases using an MMP-2- and -9-sensitive ACPP linked to Cy5 and demonstrated fluorescence labeling of tumors [130]. 

Other approaches to targeting metalloproteinases include peptides such as chlorotoxin. The molecule was discovered in the venom of the scorpion *Leiurus quinquestriatus* and binds to MMP-2, membrane type-I MMP, and transmembrane inhibitor of metalloproteinase-2. Viesh et al. used chlorotoxin conjugated to Cy5.5 to label colon cancers in a spontaneous mouse model of CRC [131]. Chlorotoxin conjugated to ICG, called BLZ-100 (also known as tozuleristide or Tumor Paint^®^), has entered clinical trials and was demonstrated to be safe and effective in visualizing breast, brain, head and neck, and skin cancers [132,133,134,135]. 

Targeting of matrix metalloproteinase is appealing, as these proteases are upregulated in a number of tumors and are not only limited to CRC. There are a number of technologies to further modify these molecules to optimize their delivery of tumor-specific fluorescence, the targeting of matrix metalloproteinase will continue to be a robust approach for FGS of CRC. 

## 12. Mucins

Mucins are high-molecular-weight epithelial glycoproteins and can either be secreted or membrane-bound. MUC2, MUC5AC, MUC5B, and MUC6 are generally accepted as being secreted, and MUC1, MUC3A, MUC3B, MUC4, MUC12, and MUC17 are membrane-bound, although there are other classes of mucins that do not fit well into either class. Mucin 1 (Muc1) has been mostly studied in colorectal cancer. It is an apical transmembrane glycoprotein that functions in cell signaling and pathogen binding [136]. MUC1 is normally expressed in glandular epithelium, where it forms a mucosal barrier. In cancer, tumor-associated Muc1 loses its apical restriction, becomes aberrantly glycosylated, and becomes overexpressed in 46–98% of CRCs, depending on the stage [137,138]. Muc1 has been targeted for tumor labeling using antibody-conjugated fluorophores and targeting peptides to image mouse model of pancreatic cancers [139,140]. Muc1-based aptamers have been used to label breast, liver, and lung tumors in vivo [141,142]. Muc1 aptamers linked to albumin particles loaded with docetaxel are being studied for targeted therapies. Preclinical evaluation of Muc1 and other mucins for FGS in CRC is currently ongoing, and the results will be promising for clinical translation. 

## 13. Vascular Endothelial Growth Factor

Vascular endothelial growth factor (VEGF) is one of the most potent angiogenic factors released by tumors to stimulate neoangiogenesis for tumor growth [143]. VEGF-A is the most studied member of the VEGF family. It is membrane-bound but predominantly secreted locally into the interstitial space [144]. It is expressed normally in endothelium, macrophages, platelets, keratinocytes, and renal epithelial cells and has physiological functions in bone formation, hematopoiesis, and wound healing [145]. VEGF is overexpressed in 50–70% of CRCs, with minimal to no expression in normal colonic mucosa [146]. VEGF is associated with metastases and poor prognosis [147]. Bevacizumab, an anti-VEGF-A antibody, is now the standard of care in combination with standard chemotherapy for primary and metastatic CRC. Angiogenesis blockade has been shown to be efficacious in CRC. There are many molecules developed in addition to bevacizumab to block VEGF signaling. These include fragmented antibodies such as ranibizumab, antibodies that bind and inhibit VEGF receptors such as ramucirumab, soluble small molecule decoys such as aflibercept, and small molecule tyrosine kinase inhibitors that inhibit VEGF in addition to other tumor-related growth factors, such as sorafenib, sunitinib, pazopanib, axitinib, and regorafenib [132]. 

Bevacizumab-based fluorescent probes are currently the most advanced VEGF-targeting molecules for FGS of CRC. Conjugation to both IRDye800 and a nuclear tracer was initially shown to be efficacious in labeling breast, gastric, and ovarian cancers in mouse models [148]. The bevacizumab–fluorophore conjugate is in clinical trials with demonstrated safety and efficacy for labeling breast cancer (NCT01508572) and peritoneal carcinomatosis of CRC origin [149,150]. It is also being used for labeling rectal cancer specimens, and the fluorescence signal can indicate positive circumferential tumor margins (Figure 3) [151]. Bevacizumab-IRDye800 is also being used for fluorescence endoscopy of rectal cancer after neoadjuvant therapy (NCT01972373) and for the detection of CRC in patients with familial adenomatous polyposis (NCT02113202) [152,153]. When applying TASC criteria to VEGF, it has a lower score primarily due to its relatively lower rate of overexpression in CRC patients and local expression in interstitial tissue. As there are many molecules that have been developed to bind VEGF and it is upregulated in a wide range of cancers besides CRC, molecular targeting of VEGF will be promising for labeling tumors. 

## 14. Cathepsins

Cathepsins are a family of proteases that function in intracellular and extracellular peptide hydrolysis. They have a physiological role in growth, development, and cellular differentiation and have been demonstrated in CRC to have a role in tumor invasion and metastasis [154]. Cathepsins, especially cathepsin B, are expressed in the cytoplasmic/nuclear space in normal colonic mucosa and switch to peripheral and basal membranes in early carcinoma [155,156]. There is especially strong expression at the tumor–stroma interface. Cathepsin B is overexpressed in over 80% of patients with CRC [157,158]. These proteases are similar to matrix metalloproteinases in probe designs for molecular targeting in that cleavable linkers are often utilized. LUM015 is the most clinically advanced of these probes and has a cathepsin B-, K-, L-, and S-activatable peptide with a fluorescence quencher and a Cy5 fluorophore [159]. LUM015 has been shown to be efficacious in animal models of CRC. It has completed phase I clinical trials [160] and has advanced to phase II clinical trials in breast cancer [161,162]. Clinical trials of LUM015 for labeling GI malignancies (NCT02584244), including CRC and peritoneal carcinomatosis (NCT03834272), are underway. Other approaches include topical application of cathepsin-activatable probes linked to a quenched fluorophore, which has shown efficacy in spontaneously mouse models of CRC. However, clinical trials using this probe have not been conducted [163]. Although LUM015 was ranked lower on the TASC criteria compared to other molecular targets at that time, it has since gained significant clinical traction in molecular fluorescence imaging of CRC. Further studies using cathepsins as tumor targeting probes for FGS will be forthcoming. 

## 15. Tumor-Associated Glycoprotein-72

Tumor-associated glycoprotein (TAG-72) is a glycoprotein found on the membrane of many adenocarcinomas, including CRC [164]. It is overexpressed in over 80% of CRCs, and serum levels correlate with the severity of disease, with only limited expression in endometrium and fetal tissue. It is estimated that over 80% of CRCs express TAG-72 [165]. It is expressed at high concentrations and is secreted by cancer cells to the tumor microenvironment, but it lacks significant serum circulation, which can decrease the background signal as a probe. Hollandsworth et al. used a clinically translatable version of an anti-TAG-72 antibody that had been humanized and linked to IRDye800 and demonstrated the detection of CRC less than 1 mm in size [166]. Gong et al. demonstrated a single-chain variable fragment of an anti-TAG-72 antibody linked to IRDye800 and showed rapid tumor labeling within 3 h after IP injection in an orthotopic mouse model of CRC [167]. Cohen et al. used topical application of anti-TAG-72-conjugated NIR fluorescent albumin nanoparticles for labeling orthotopic mouse colon cancer, with optimal imaging occurring after 4 h [167]. There are currently no clinical trials using TAG-72 for fluorescence labeling, but as it is a probe ideal for molecular labeling, further studies will be promising. 

## 16. Folate Receptor Alpha 

Folate receptor alpha (FRα) is a membrane-bound protein that is overexpressed in a number of cancers [168]. It is hypothesized that cancer cells upregulate these receptors due to increased requirements for folate, which is necessary for increased synthesis of nucleotide bases needed for cell proliferation [169]. FRα is overexpressed in 30–33% of CRCs and has limited expression in normal tissue [170,171]. Because of this low percentage in patients with CRC, FRα received a lower TASC criteria score. However, fluorescence-conjugated folic acid has been discovered to be an optimal targeting ligand of upregulated folate receptors, due to ease of conjugation and high affinity, and this has been exploited for imaging and therapeutic applications [172]. Van Dam et al. showed the clinical feasibility of folic acid conjugated to fluorescein isothiocyanate (EC17) in the first-in-human FGS using a molecularly targeted agent during cytoreductive surgery for ovarian cancer [173]. As fluorescein isothiocyanate is a visible-wavelength fluorophore and has poor tissue penetration, conjugation with a NIR fluorophore was performed to create OTL38. The advantages of this probe are rapid labeling, stability, and ease of production [172]. The safety and efficacy of OTL38 were shown in phase I/II clinical trials in ovarian cancer (NCT02317705), lung cancers (NCT02872701), and renal cell cancers (NCT02645409), and it is currently undergoing phase III clinical trials. OLD38 should be an ideal agent for targeted probe in CRC patients with overexpression of FRα.

## 17. Conclusions

The development of molecularly targeted probes for FGS of CRC has made remarkable advances in recent years. There are now even wider arrays of targeting platforms and targets that have been developed. Many of these probes for tumor labeling have advanced from preclinical proof-of-principle studies to phase I/II and even phase III clinical trials for tumor detection. While antibodies initially dominated in the field, other molecules have since advanced, and more will emerge for tumor visualization to improve cancer-related outcomes, especially in rectal cancer and peritoneal carcinomatosis of CRC origin. 

Other applications of FGS that will be forthcoming are fluorescence-enhanced endoscopic techniques, many of which are concurrently being evaluated with fluorescent molecular probes and fluorescence-capable endoscopes [36,152]. This could assist in the more accurate detection of CRC in high-risk patients in whom endoscopic topography is difficult due to inflammatory bowel disease [174]. There will be further studies using this technology for organ-sparing approaches such as endoscopic submucosal dissection or in full-thickness resections of early-stage tumors, as well as tumors that have had significant neoadjuvant treatment. Fluorescent tumor labeling of patients who had significant neoadjuvant treatment is currently underway in the Netherlands, with preliminary studies by Tjalma et al. showing that visualization using bevacizumab-IRDye800 can distinguish residual tumor from normal rectal tissue and fibrosis. Bevacizumab-IRDye800 improved the prediction of final pathologic results in 16% of patients compared to standard MRI and white-light endoscopy [153]. Fluorescence labeling can also be incorporated into the decision-making process of patients with CRC who are eligible for the “watch-and-wait approach” to further improve outcomes and intervene at the appropriate time with salvage surgery. 

Future areas that need to be explored are evaluations on the fidelity of tumor-specific molecular labeling in tumors with variable expression of the target. What is the threshold needed for effective labeling if there is heterogeneous expression? Current preclinical models used to validate candidate probes utilize tumors with a homogeneous population of cells expressing the target, but they do not account for a mixed population [175]. Evaluation of the probe in patient-derived xenografts or spontaneous carcinogenesis models can facilitate clinical translation [176]. Neoadjuvant therapies can change the tissue expression of the target and change the predominant cell population over time, further compounding this problem. A cocktail of tumor-targeting probes would ideally be used to further optimize tissue selectivity, but this approach may be limited by cost [175,177]. 

With more agents entering clinical trials, ideal probe selection will become an issue. Should probe selection be driven by labeling from preoperative biopsy samples or from serum tests? If the target is then secreted in the serum, as is the case with CEA, how will this affect imaging, and should doses be modified? What should be the approach if a preoperative tissue diagnosis is unavailable or too resource-intensive and morbid to obtain? In tumors such as brain cancer, preoperative biopsies are rarely performed. Are there more noninvasive ways to personalize selection of the tumor labeling probe? Information from a serum molecular profiling of circulating tumor DNA could reveal genomic and proteomic level information. Other noninvasive approaches could include targeted radionuclide scans, such as the DOTA-TATE scan for neuroendocrine tumors or prostate-specific membrane antigen (PSMA) scan for prostate cancer; however, a molecularly targeted scan for CRC does not yet exist. 

As fluorescent molecular probes are becoming more sophisticated, the development of imaging devices will need to follow suit. A current limitation of macroscopic imaging is that the detection of bulk tumor contrast is less sensitive compared to microscopic contrast. The lack of a clinically detectable fluorescence signal may not be indicative of the specificity of the molecule but rather a limitation of the sensitivity of the detection device. However, very high sensitivities of detection could lead to a false positive fluorescence signal. NIR imaging devices are not standardized and it is difficult to compare among device manufacturers. While it is simple to compare the resolution of cameras, it is difficult to quantify device performance in relation to tissue optical properties. These issues of quality control and standardization will need to be addressed as the field of tumor-labeling and FGS matures. 

The use of FGS for CRC is advancing and enthusiasm for the adoption of targeted fluorescence molecular imaging in surgery is increasing. There remain challenges and many questions that need to be answered to further optimize the use of these agents. Further work will be forthcoming, as this is a very active area of translational research. The potential to more accurately diagnose, detect, and resect CRC can be enhanced by fluorescence imaging. 

## Figures and Tables

**Figure 1 cells-11-00249-f001:**
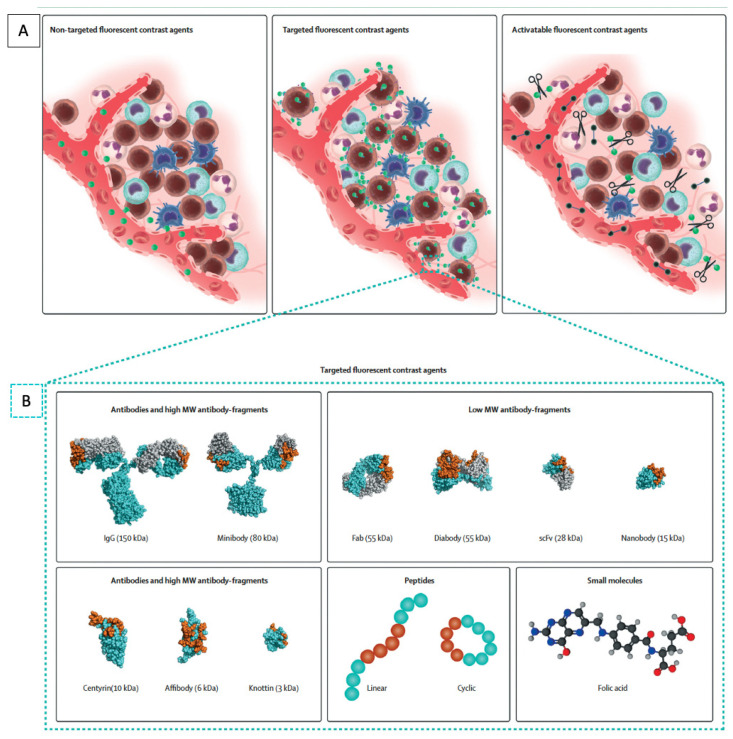
Molecular fluorescent contrast agents and targeting moieties used for intraoperative imaging during cancer surgery. (**A**) Schematic representation of the mode of action of the different types of fluorescent contrast agents. Non-targeted fluorescent contrast agents such as indocyanine green passively accumulate in tumor tissue via the enhanced permeability and retention effect. Targeted fluorescent contrast agents, consisting of a fluorescent dye conjugated to a targeting moiety, actively accumulate in tumor tissue by recognizing a specific biomarker expressed by tumor cells or tumor-associated stromal cells. Imaging is performed once unbound tracers have been cleared sufficiently. Activatable fluorescent contrast agents remain optically silent until fluorescent dyes are released by enzymatic digestion of their backbone. (**B**) Schematic representation of the different classes and subclasses of targeting moieties used for the design of targeted fluorescent contrast agents: antibodies, antibody fragments, protein scaffolds, peptides, and small molecules. Representative space-filling images of an antibody (1IGT), Fab fragment (6B9Z), diabody (1MOE), scFv (1P4I), nanobody (5MY6), centyrin (5L2H), affibody (2KZJ), and knottin (2N8B) were obtained from the RCSB protein bank and prepared using PyMOL. The space-filling minibody model is an interpretation created using PyMOL; antigen-binding regions are highlighted in orange. Adapted from Hernot et al. [48].

**Figure 2 cells-11-00249-f002:**
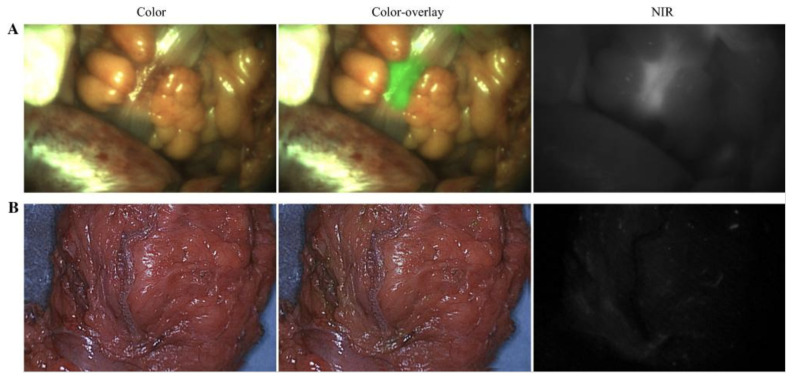
Example of true-positive and true-negative fluorescence signals in patients with colorectal carcinoma using SGM101, an anti-CEA antibody-based tumor-targeting probe. (**A**) Intraoperative fluorescence of a palpable colorectal tumor during surgery, with a TBR of 2.0 (true positive). (**B**) Absence of fluorescence in a tumor, which was confirmed as a pathological complete response by histopathology (true negative). TBR tumor-to-background ratio, NIR near-infrared. Adapted from de Valk et al. [77].

**Figure 3 cells-11-00249-f003:**
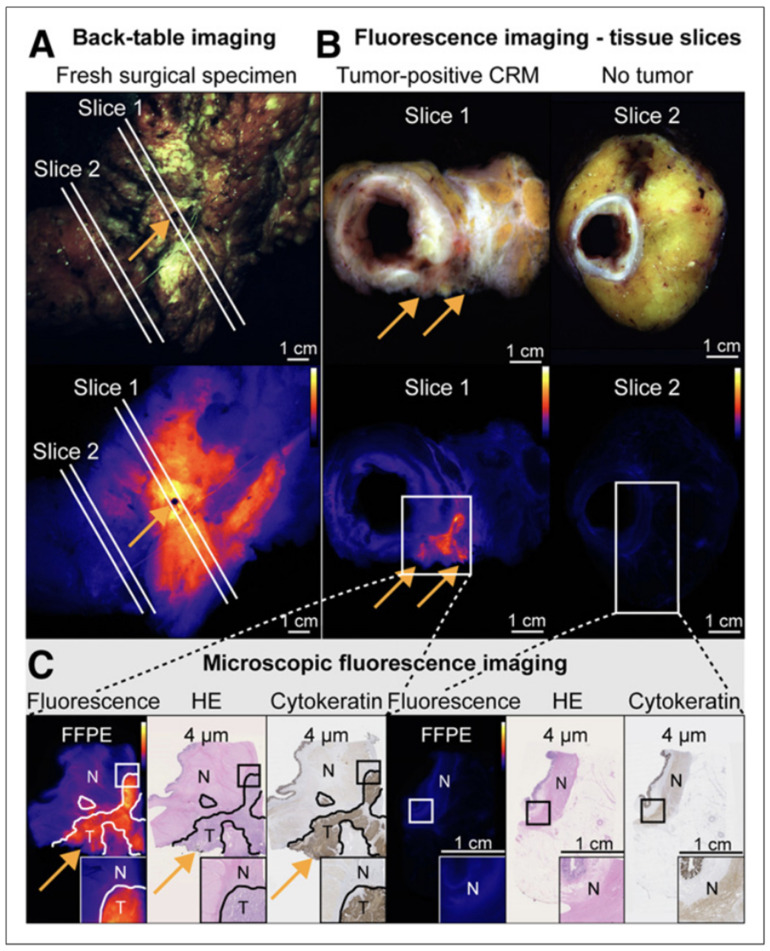
Back-table fluorescent tumor labeling of patient with tumor-positive circumferential resection margin (CRM) using bevacizumab-IRDye800. A black pin at the location shows increased fluorescence, to enable accurate correlation with histology ((**A**), orange arrow). Fluorescence imaging of 2 corresponding tissue slices (**B**) and further microscopic fluorescence imaging and histologic correlation (**C**), with orange arrows indicating the location of tumor-positive CRM. High fluorescence signals were observed at CRM of tissue slice 1, containing tumor-positive CRM, whereas low fluorescence signals were observed in nontumor tissue slice 2, corresponding to microscopy results. FFPE: formalin-fixed paraffin-embedded; HE: hematoxylin–eosin; N: nontumor; T: tumor. Adapted from de Jongh et al. [151].

## Data Availability

Not applicable.

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
