# Peer review of "Fluorescence Molecular Targeting of Colon Cancer to Visualize the Invisible"

_cells, 2022, doi:10.3390/cells11020249_

Round 1
Reviewer 1 Report
The paper titled "Fluorescence molecular targeting of colon cancer to visualize
the invisible" investigate an area of interest in colorectal surgery in recent years. It represents a comprehensive review of currently available methods to use fluorescence in colorectal surgery.
I only have some minor suggestions:
- the introduction paragraph and the abstract are identical. Please expand the introduction producing more evidence about the background. I would not focus on the use of fluorescence to obtain an R0 margin, as this should be routinely achieved through good-quality surgery.
- I would not state that "Traditional approaches to surgery OFTEN leave positive margins which are negatively associated with outcome." It is true that positive margins are associated with negative outcomes, however that does not happen OFTEN.
Author Response
We appreciate the careful consideration of our manuscript for publication in Cells. We welcome the opportunity to improve our manuscript. In the following document, we have addressed each of the concerns and have made modification to the manuscript and figures accordingly.
In our revised manuscript, we have addressed all points raised by the reviewer, and believe that our responses will satisfy their concerns. Point-by-point responses are given below.
Reviewer 1:
The paper titled "Fluorescence molecular targeting of colon cancer to visualize
the invisible" investigate an area of interest in colorectal surgery in recent years. It represents a comprehensive review of currently available methods to use fluorescence in colorectal surgery.
I only have some minor suggestions:
- the introduction paragraph and the abstract are identical. Please expand the introduction producing more evidence about the background. I would not focus on the use of fluorescence to obtain an R0 margin, as this should be routinely achieved through good-quality surgery.
- The introduction and the abstract have been revised in the manuscript with the abstract further summarizing the remainder of the article and the introduction giving further background on the topic at hand. Detailed evidence is further described in the immediate section following the introduction.
- I would not state that "Traditional approaches to surgery OFTEN leave positive margins which are negatively associated with outcome." It is true that positive margins are associated with negative outcomes, however that does not happen OFTEN.
- We agree with the reviewer and have revised the manuscript to reflect this. The section now states “. Traditional principles to localization and confirmation of complete resection during surgery are: pre-operative cross-sectional imaging, identification of anatomic boundaries, palpation of the lesion, clinical judgment, and frozen sections if degree of suspicion is high. However, this traditional approach can be subject to detection and sampling errors, which can leave behind positive margins.”
Reviewer 2 Report
Lwin and coworkers review the current status of fluorescently labeled agents to help surgical operations on CRC patients. The topic is of broad interest and well organized. I recommend acceptance with only minor adjustments, detailed in what follows.
- Section4, second paragraph, second sentence
it *is* orally administered - The authors do not mention the use of fluorescent nanoparticles as fluorescent probes (Section 4).
- Some abbreviations are not declared in the main text, e.g. npv. In the caption of figure 1, abbreviation explanation is after the abbreviation is actually being used
- Can the author comments what happens with those targeted dyes that only represent a fraction of cancer cell populations? It is known that population hyerarchies exist in CRC, and some of the proposed markers only mark a subset of cells. Is the resolution of endoscopy enough to appreciated such changes particulary in considering the margins vs the bulk of the tumor?
- is it feasible (now or in the future) to combine more than one target? As the authors correctly report, some of these markers are not expressed in all tumors, and combination (possibly even with one common fluorophore) might be an option to improve the contrast
- is the choice of the targeted dyes based on some kind of personalized assay? Is it conceivable to use liquid biopsy or other non invasive techiques to pick the best possible fluorescent targeted agent?
- the discussion is a bit short and lacks for the necessary future directions, both technological and clinical. Some of my comments might be discussed to make the discussion a bit more open towards possible improvements.
- A sentence in the discussion: "as fluorescent probes design can be applicable..." makes little sense.
- A few sentences could be dedicated to let the reader understand how is the imaging actually employed during or post-surgery.
Author Response
We appreciate the careful consideration of our manuscript for publication in Cells. We welcome the opportunity to improve our manuscript. In the following document, we have addressed each of the concerns and have made modification to the manuscript and figures accordingly.
In our revised manuscript, we have addressed all points raised by the reviewer, and believe that our responses will satisfy their concerns. Point-by-point responses are given below.
Reviewer 2:
Lwin and coworkers review the current status of fluorescently labeled agents to help surgical operations on CRC patients. The topic is of broad interest and well organized. I recommend acceptance with only minor adjustments, detailed in what follows.
- Section4, second paragraph, second sentence
it *is* orally administered - The manuscript has been revised to reflect this. The sentence now states “It is orally administered and has been used widely in neurosurgery”
- The authors do not mention the use of fluorescent nanoparticles as fluorescent probes (Section 4).
- The manuscript has been revised to reflect this discussion on nanoparticles: “Other classes of tumor-targeting molecules are peptides and nanoparticles….Nanoparticles are very small molecules, usually 1-100 nanometers in diameter, with efficient cell penetration. They are highly versatile structures to carry bioactive molecules and can be linked a tumor-targeting moiety and fluorescence [61].
- Additionally, discussions on CEA and Tag72 based nanoparticles for in-vivo fluorescence labeling were added to corresponding sections. “Silica nanoparticles linked to anti-CEA antibodies have been evaluated in CRC subcutaneous tumor models and showed tumor-specific localization after 6 hours [80]. The advantage of nanoparticles care that they can potentially overcome many of the traditional limitations compared to traditional antibody-fluorophore conjugation such as the improved emission brightness, quantum yield, and photo-stability [81]” “Cohen et al used a topical application of an anti-TAG-72 conjugated NIR fluorescent albumin nanoparticles for labeling of orthotopic mouse colon cancer with optimal imaging after 4 hours [167].”
- Some abbreviations are not declared in the main text, e.g. npv. In the caption of figure 1, abbreviation explanation is after the abbreviation is actually being used
- The manuscript has been revised to reflect this. The sentence now states: “It is noteworthy that with a high negative predictive value (NPV)…”
- Can the author comments what happens with those targeted dyes that only represent a fraction of cancer cell populations? It is known that population hierarchies exist in CRC, and some of the proposed markers only mark a subset of cells. Is the resolution of endoscopy enough to appreciated such changes particulary in considering the margins vs the bulk of the tumor?
- An additional paragraph under “Conclusions” has been added to address this issue: “Future areas of exploration that needs to be performed in this area are evaluations on the fidelity of tumor-specific molecular labeling in tumors with variable expression of the target. What is the threshold needed for effective labeling if there is heterogeneity of expression? Current pre-clinical models used to validate candidate probes utilize animal models with a homogenous population of cells expressing the target, but do not account for a mixed population. Evaluation of the probe in patient-derived xenografts or spontaneous carcinogenesis models can help bridge this gap [175].”
- is it feasible (now or in the future) to combine more than one target? As the authors correctly report, some of these markers are not expressed in all tumors, and combination (possibly even with one common fluorophore) might be an option to improve the contrast
- An additional paragraph under “Conclusions” has been added to address this issue: “Neoadjuvant therapies can change tissue expression of the target and change the predominant cell population over time, further compounding this problem. A cocktail of molecularly targeted agents could ideally be used to further optimize tissue selectivity, but this approach may be limited by cost [176,177].”
- is the choice of the targeted dyes based on some kind of personalized assay? Is it conceivable to use liquid biopsy or other non invasive techiques to pick the best possible fluorescent targeted agent?
- An additional paragraph under “Conclusions” has been added further discussing this issue: “With more agents entering clinical trials, ideal probe selection will become an issue. Should probe selection be driven by staining from pre-operative biopsy samples or from serum tests? If the target is then secreted in the serum, as is the case with CEA, how will this affect imaging, and should doses be modified? What should be the approach if a pre-operative tissue diagnosis is unavailable or too resource intensive and morbid to obtain? In tumors such as brain tumors, where pre-operative biopsies are rarely performed. Are there more non-invasive ways to personalize the intra-operative molecular labeling agent? Information from a serum molecular profiling of circulating tumor DNA could reveal information on a genomic and proteomic level. Other non-invasive approach could include targeted radionuclide scans such as the DOTA-TATE scan for neuroendocrine tumors or prostate-specific membrane antigen (PSMA) scan for prostate cancer, however a molecularly targeted scan for CRC does not yet exist..”
- the discussion is a bit short and lacks for the necessary future directions, both technological and clinical. Some of my comments might be discussed to make the discussion a bit more open towards possible improvements.
- We appreciate the thoughtful comments and suggestions and the “Conclusions” section has been revised with the great discussion points.
- A sentence in the discussion: "as fluorescent probes design can be applicable..." makes little sense.
- The manuscript has been revised to reflect this and the sentence removed.
- A few sentences could be dedicated to let the reader understand how is the imaging actually employed during or post-surgery.
- An additional paragraph under “Impact of margin positivity on colorectal cancer and the potential of fluorescence guidance” has been added to the manuscript to better describe this: “Intraoperatively, fluorescence imaging can performed as a diagnostic laparoscopy to detect the primary tumor and or peritoneal/liver metastases. Once the surgery is underway, fluorescence imaging can be utilized periodically to better define the location of the tumor in relation to surrounding tissue and anatomic structures. This can be performed in a minimally-invasive fashion with fluorescent capable laparoscopes, or in a traditional open laparotomy with hand-held devices. Fluorescence imaging can potentially be used to detect lymphatic drainage, although this can often be impaired due to the depth of overlying tissue in patients with obesity and thickened fatty mesentery. After the specimen is removed, fluorescence imaging can be performed on both the surgical bed and the specimen to ensure complete tumor removal and an adequate rim of normal tissue or mesorectal envelope.